# Production of Protein Hydrolysate from Quinoa (*Chenopodium quinoa* Willd.): Economic and Experimental Evaluation of Two Pretreatments Using Supercritical Fluids’ Extraction and Conventional Solvent Extraction [note 1]

**DOI:** 10.3390/foods11071015

**Published:** 2022-03-30

**Authors:** Luis Olivera-Montenegro, Alejandra Bugarin, Alejandro Marzano, Ivan Best, Giovani L. Zabot, Hugo Romero

**Affiliations:** 1Grupo de Ciencia, Tecnología e Innovación en Alimentos, Universidad San Ignacio de Loyola, Lima 15024, Peru; abugarin@usil.edu.pe (A.B.); lmarzano@usil.edu.pe (A.M.); ibest@usil.edu.pe (I.B.); 2Laboratory of Agroindustrial Processes Engineering (LAPE), Federal University of Santa Maria (UFSM), 1040 Sete de Setembro St., Center DC, Cachoeira do Sul 96508-010, RS, Brazil; giovani.zabot@ufsm.br; 3Electroanalytical Applications and Bioenergy Research Group, Chemical Engineering Department, Technical University of Machala, Av. Panamericana Km 5 ½, Machala 070102, Ecuador; hromero@utmachala.edu.ec

**Keywords:** quinoa oil, bioactive peptides, SFE, economic evaluation, quinoa protein hydrolysate

## Abstract

Supercritical fluids’ extraction (SFE) and conventional solvent extraction (CSE) for defatting of quinoa flour as pretreatments to produce the quinoa protein hydrolysate (QPH) were studied. The objective was to extract the oil and separate the phenolic compounds (PC) and the defatted quinoa flour for subsequent quinoa protein extraction and enzymatic hydrolysis. The oil extraction yield (OEY), total flavonoid content (TFC), and QPH yield were compared. SuperPro Designer 9.0^®^ software was used to estimate the cost of manufacturing (COM), productivity, and net present value (NPV) on laboratory and industrial scales. SFE allows higher OEY and separation of PC. The SFE oil showed a higher OEY (99.70%), higher antioxidant activity (34.28 mg GAE/100 g), higher QPH yield (197.12%), lower COM (US$ 90.10/kg), and higher NPV (US$ 205,006,000) as compared to CSE (with 77.59%, 160.52%, US$ 109.29/kg, and US$ 28,159,000, respectively). The sensitivity analysis showed that the sale of by-products improves the economic results: at the industrial scale, no significant differences were found, and both processes are economically feasible. However, results indicate that SFE allows the recovery of an oil and QPH of better nutritional quality and a high level of purity-free organic solvents for further health and nutraceutical uses.

## 1. Introduction

Quinoa (*Chenopodium quinoa* Willd.) is a pseudocereal that has been grown in the Andes region for more than 5000 years [1]. It has a mass protein content of 10–20%, which can vary according to the phenotype [2]. Quinoa has most of the essential amino acids required by the human body, such as lysine, histidine, and methionine, which are generally the limiting amino acids in common cereals [3]. Quinoa has a considerable oil content, depending on the varieties, with the following reported values (g oil/g insoluble solid) for four varieties: Titikaka (4.3 ± 0.1), Pasankalla (7.9 ± 0.1), Altiplano (6.0 ± 0.4), and Collana (4.9 ± 0.3) [4]. Quinoa also has bioactive compounds such as phenolic compounds, demonstrating a high potential for use in the food industry as a functional ingredient, or as a nutraceutical or replacement of synthetic antioxidants. Carciochi et al. reported the extraction of phenolic compounds using ethanol, obtaining extracts with a total phenolic content ranging from 67.50 to 102.86 mg GAE/100 g dry basis and an antioxidant activity of 28.9% DPPH radical scavenging [5]. These bioactive compounds can cause interferences in the analysis of the antioxidant activity of the quinoa protein hydrolysate, because phenolic compounds have biological activities; thus, it is recommended to separate these compounds from food matrixes by clean technologies such as supercritical fluid extraction or ultrasound before proceeding with enzymatic hydrolysis [6].

Peru is currently one of the world’s leading quinoa exporters, surpassing Bolivia since 2014. Additionally, based on information on the presentation of exported products, published by Peru Trade Now (PROMPERU, 2020), round 91% in FOB value exported is in the form of grain, 3.6% in flakes, and 2.2% in flour [7]. Worldwide, quinoa cultivation has increased from 8 countries in 1980 to 100 by 2021 [8]. In this sense, the functional food market is constantly growing at a global level, which was estimated at US$ 162 billion in 2018 and was projected to reach US$ 280 billion by 2025, with an annual growth rate of around 8% [9]. This reaffirms the importance of focusing efforts on the industrialization of functional foods and nutraceuticals. The COVID-19 pandemic has increased the size of the global bioactive compounds and food ingredients market, and among them, some bioactive peptides may have health benefits and a possible action mode against SARS-CoV-2 [10].

Protein hydrolysates and peptide fractions with determined molecular mass or peptides can be used as functional foods, nutraceuticals, or additives to food products, increasing their nutraceutical potential. Good sources of peptides are protein-rich food products of plant, animal, or alternative origin [11]. The quinoa protein is a good precursor of bioactive peptides, and of the biological functionalities of quinoa protein hydrolysate, including is antioxidant activity [12]. From the above, alkaline extraction and isoelectric precipitation are the most common techniques used for quinoa protein extraction [13]. Before the extraction of protein, the grounded seeds are treated with organic solvents such as petroleum ether, methanol, chloroform, and hexane to defatting the quinoa flour [11,12,13]. However, there is little research on the use of supercritical fluids’ for the defatting of quinoa flour. Benito-Román et al. and Wejnerowska and Ciaciuch used supercritical CO_2_ and ethanol as cosolvent for the defatting of quinoa flour [4,14]. The advantages of using supercritical CO_2_ over conventional methods are many: supercritical carbon dioxide (CO_2_) is the most widely used solvent because it is non-toxic, non-explosive, inexpensive, easy to separate, and allows selective extractions by varying the pressure and temperature [15].

Biochemical methods are not feasible to produce the peptides at the industrial level with a higher yield and low cost. In this regard, novel processing technologies are required as promising technologies that can possibly be coupled with biochemical hydrolysis methods to produce antioxidant peptides with better yield and bioactivities in a shorter time and lower cost than biochemical methods [13]. Further research is required for industrial production of bioactive quinoa peptides [16]. In general, it is necessary to increase the advancement of processing technologies, due to people’s increased awareness of the importance of bioactive peptides as health-promoting ingredients [17]. Previous research shows the technological and economic evaluation of the production of quinoa (*Chenopodium quinoa* Willd.) hydrolyzed using supercritical fluids’ and conventional extraction with organic solvents, as a pretreatment to separate the fat and phenolic compounds in a single extraction process, with the purpose of producing protein hydrolysates with a source of antioxidant bioactive peptides [18]. Therefore, the objective of this work was to compare the oil extraction yield, total flavonoid content, and quinoa protein hydrolyzed yield. An economic evaluation and sensitivity analysis were then carried out using SuperPro Designer^®^ 9.0 software (Intelligen Inc., Scotch Plains, NJ, USA), and the scale-up was with a volume of 1.5 kg (Laboratory) up to 2500 kg (Industrial).

## 2. Materials and Methods

### 2.1. Sample Preparation

Quinoa (*Chenopodium quinoa* Willd.) used in this work was supplied in March 2018 by the Cereals Programme of the National Agrarian La Molina, IRD Saint Joan de Yanamuclo, Junin Region, Peru (latitude: 11°51′37.5″ S, longitude: 75°23′52.0″ W). The variety used was Huallhuas (INIA-415) for reasons of variety identification, standardization, and uniformity of grain quality. The grains were washed with distilled water at approximately 20 °C to remove saponins, then were milled into a powder to obtain particle sizes ≤0.5 mm (for subsequent assays, as previously described [19].

### 2.2. Conventional Solvent Extraction and Supercritical Fluids’ Extraction Process for Defatted Quinoa Flour

Two extraction technologies were evaluated for the defatted quinoa flour, as a stage prior to enzymatic hydrolysis and obtaining quinoa protein hydrolysate. The first technology used was conventional solvent extraction (CSE) to extract the oil, by petroleum ether. The solid/solvent ratio parameters were 1:3.33 for 19 h at 4 °C, according to a process developed by Olivera-Montenegro et al. [19], based on the methodology of Fritz et al. [20] for obtaining protein hydrolysates from amaranth, with slight modifications. For each extraction, the 2 L extraction vessel was filled with approximately 300 g of quinoa flour with 1000 mL of petroleum ether. The second technology used was supercritical fluids’ extraction (SFE) for the recovery of oil and total phenolic compounds (lipid fraction), and multi-solvent equipment was used for pretreatment (2802.0000 Top Industry, Vaux-le-Pénil, France). The equipment has the following dimensions: 2.40 × 2.40 × 0.85 m in (L × H × T), and the extractor has a volume of 1 L. This equipment is designed to perform different extraction techniques, which includes supercritical CO_2_ extraction with or without a polarity modifier. The extractor equipment has a cosolvent pump that flows the ethanol from a flask to the reactor containing the sample. This technique was applied for the extraction of oil and phenolic compounds from quinoa flour. Operational parameters were: T_reactor_ = 55 °C, P = 23 MPa, and 8 g of quinoa/100 mL ethanol. All runs were at CO_2_ mass flow = 35 g/min with an extraction time of 4 h, and the experimental procedure at the laboratory scale was previously described [19].

The oil extraction yield (OEY) was calculated for both extraction processes as the ratio between the total mass of extract and the mass of raw material loaded in the extractor on a dry weight (dw) basis.

The hydrolysis of QPH slurry was carried out following the methodology described by Olivera-Montenegro et al. [19]. Briefly, a 2.5% (*w*/*v*, protein basis) slurry of the quinoa protein concentrate was prepared in distilled water and the pH was adjusted to 7.0 with 0.2 M phosphate buffer. Based on the protein content of the concentrate and the enzyme dose, the final formula considered in both processes at the laboratory and industrial scales was: 3.47% of protein concentrate, 96.51% of solution, and 0.02% of enzyme, and a 10% decrease in enzyme inactivation was considered due to evaporation losses at the industrial scale. Finally, QPH yield was determined as the ratio of the total mass of hydrolysate to the mass of the protein in the starting material on a dry weight (dw) basis, adapted from a previous study [21].

### 2.3. Total Phenolic Content and Total Flavonoid Content Determination

Total phenolic content (TPC) in the oil extracted was estimated using the Folin–Ciocalteau reagent method, as described by Olivera-Montenegro et al. [19,22]: 100 µL of the oil sample was mixed with 750 µL of the Folin–Ciocalteu reagent, 0.2 N, and 5 min after the reaction, 750 µL of anhydrous sodium carbonate at 7.5% was added. The reaction was carried out at 25 °C in darkness for 30 min. The absorbance was measured at 725 nm. The calibration curve was constructed using gallic acid as a reference standard. The results were expressed as gallic acid equivalents (GAE) in mg/100 g of quinoa seeds, and flour before and after treatment with SFE on a dry weight (dw) basis.

The total flavonoid content (TFC) in QPH (remaining flavonoid) was measured using the colorimetric method, as reported by Sarket et al. [23]. Briefly, 1.5 mL of methanol was added to 0.1 mL of 10% aluminum chloride, 0.1 mL of 1 M potassium acetate, 2.8 mL of distilled water and 500 µL of quinoa protein hydrolysate for 30 min at 20 °C. Absorbance was then measured at 415 nm using a spectrophotometer. TFC was expressed as μg rutin equivalent/g of dry matter.

### 2.4. Process Simulation Model

Simulations of the SFE and CSE processes were performed using the SuperPro Designer 9.0^®^ software (Intelligen Inc., Scotch Plains, NJ, USA), in which direct costs consist of raw material costs, operational labor, utilities (electricity, steam, treated water, cooling and heating system), and facilities, among others. Indirect costs, including taxes, depreciation, insurance, business maintenance, and other administrative expenses, are also calculated by the software [24]. The process model was build using parameters and process conditions obtained from Olivera-Montenegro et al. [19], for both processes (SFE and CSE), as presented in Table 1.

Flowsheets of the SFE and CSE processes are shown in Figure 1. It was observed that the yield obtained at the industrial scale during the scale-up process can increase the QPH yield compared to the laboratory scale, under the same process conditions in both technologies (pressure, temperature, extraction time, density) [26].

Figure 2 and Figure 3 show industrial operations Gantt charts, illustrating the starting and finishing of an operation in the process of production of QPH using SFE and CSE, respectively. The charts consider a plant for processing 2500 kg of quinoa seeds per day. Operating times for both processes were based on the S/F mass ratios (washing to separate saponins, defatting, protein extraction, and enzymatic hydrolysis). The SFE process includes: a wash tank (P-1/WSH-101), a decanter (P-02/DS-103), a fluid bed dryer (P-03/FBDR-101), a hammer mill (P-04/GR-101), a sieve (P-05/VSCR-101), the quinoa flour is placed in the supercritical CO_2_ unit (P-06/R-101, P-16/MX-103, P-12/EC-101, P-13/GP-102, P-25/MX-102, P-9/GP-101, P-7/U-101, P-10/FSP-101, P-11/G-101, P-8/C-101), the defatted quinoa is placed in the stirred tank reactor (P-14/MX101, P-15/MSX-101), a decanter (P-17/DE-101), a filter press (P-18/RVF-101), the supernatant is placed in the stirred tank reactor (P-19/V-102), a washing tank (P-20/WHS-102), a decanter (P-21/DS-102), the quinoa protein is placed in the industrial spray dryer (P22/SDR-101), the dry quinoa protein is placed in the stirred tank reactor for enzymatic hydrolysis (P-23/BR-101), and a decanter (P-24/DS-101), until obtaining the quinoa protein hydrolysate.

On the other hand, the CSE process includes: a wash tank (P-1/WSH-101), a decanter (P-02/DS-103), a fluid bed dryer (P-03/FBDR-101), a hammer mill (P-04/GR-101), a sieve (P-05/VSCR-101), the quinoa flour is placed in the stirred tank reactor (P-09/MX-102, P-06/SMSX-101), the defatted quinoa is placed in the stirred tank reactor with a cooling system (P-14/MX101, P-15/MSX-101), a decanter (P-17/DE-101), a filter press (P-18/RVF-101), the supernatant is placed in the stirred tank reactor (P-19/V-102), a washing tank (P-20/WHS-102), a decanter (P-21/DS-102), the quinoa protein is placed in the industrial spray dryer (P22/SDR-101), the dry quinoa protein is placed in the stirred tank reactor for enzymatic hydrolysis (P-23/BR-101), and a decanter (P-24/DS-101), until obtaining the quinoa protein hydrolysate.

### 2.5. Scale-Up and Economic Evaluation of QPH Production

Equation (1) was used to scale-up the cost of equipment to the required capacity, where *C*_1_ is the cost of equipment with capacity *Q*_1_; in the same way, *C*_2_ is the known base cost for equipment with capacity *Q*_2_ and *n* is a constant specific to each equipment, which was obtained from [27,28].
(1)$$C_{1}=C_{2}\;{\left(\frac{Q_{1}}{Q_{2}}\right)}^{n}$$

The base costs obtained by local quotation, including import fees for the items not produced in Peru, were acquired in 2019. Unit base costs and *n* values used for the QPH plant for the SFE and CSE processes are presented in Table 2. COM can be determined as the sum of the three main components: direct costs, fixed costs, and general expenses, and was estimated according to the methodology proposed by Turton et al. [28] by using Equation (2). According to Equation (2), the three main components are estimated in terms of five operational major costs: fixed capital investment (FCI), cost of raw material (CRM), cost of labor (COL), cost of utilities (CUT), and cost of waste treatment (CWT). Table 3 shows the economic parameters to determine the COM.
COM = 0.304 × FCI + 2.73 × COL + 1.23 × (CUT + CWT + CRM)(2)

The FCI includes expenses related to the implementation of the production line (extraction plants and complementary equipment). COL is related to the number of operators of the extraction units and other process operations. CUT involves the energy used in the solvent cycle for steam generation, heating, refrigeration, and electricity requirements. CRM includes the raw material and solvent costs. Finally, the CWT of SFE was considered zero because the waste generated by the process can be considered non-hazardous, clean, and can be reused in other applications or just disposed of as ordinary plant waste, except for in the CSE process, which considered the disposal of the recovered organic solvents [29]. Direct costs including buildings, electrical facilities, yard improvement, insulation, instrumentation, installation, etc., and indirect costs including engineering, construction, administration rates, insurance, human resources for administration, cleaning services, etc., were estimated by the simulator and considered in the economic evaluation. These direct costs and FCI represent the total capital investment (TCI). Considering these aspects, more information is provided on the data used for the COM simulation (Table 3).

At the laboratory and industrial scales, a daily raw material processing capacity of 1.5 and 2500 kg was considered, respectively. This mass flow rate was used to determine the volume of the extraction vessel with SFE, using the bulk density of quinoa flour for a particle size ≤ 0.55 mm, which was reported by Cotovanu et al. as 0.55 g/mL [25], and a reduction of 14% was achieved until the quinoa flour was obtained, with the volume of the extraction vessel being 2.35 L for the laboratory scale and 4000 L for the industrial scale. Additionally, for the calculation of the SFE fluid extraction unit, Rocha-Uribe et al. [30] report an experimentally determined equation, which relates the volume of the extraction unit in m^3^ and the cost of the equipment of the whole extraction unit at the industrial scale. The equation is shown below:Y = 31,901x^0.6909^(3)

Based on the application of Equation (3), the cost of the unit at the industrial scale was determined to be US$ 9,828,054.25 (see Table 2). For the volumes of the CSE for defatting, it was considered to use a reactor tank with agitation, whose volume is 12 L (laboratory) and 15,000 L (industrial plant); in both processes, the volumes were calculated based on the bulk density (0.55 g/mL) of the quinoa flour to be processed and the density of petroleum ether (0.65 g/mL).

For the scale-up for both processes, the plant was designed to run three daily shifts for 330 days/year, equivalent to 7920 h/year. A laboratory scale of 1.5 kg/batch of quinoa seeds and an industrial scale of 2500 kg/batch quinoa seeds were considered. In the defatted stage, the flow diagrams obtained with the simulator for both processes were as follows. The CRM consist mainly of the cost of quinoa (*Chenopodium quinoa* Willd.), which was quoted at US$ 1567/ton (direct quotation of wholesale market, Lima, Peru, in 2021). The commercialization of QPH obtained by the SFE and CSE was estimated at US$ 200.00/kg. The price of oil was estimated at US$ 33.55/L and saponins was estimated at US$ 30.75/kg. Fixed capital investment (FCI) and the other input information was calculated for both plants at a production scale of 2500 kg/batch, as shown in Table 3.

### 2.6. Sensitivity Analysis

Simulation of the scale-up was carried out from a volume of 1.5 kg (laboratory) up to 2500 kg (industrial plant). The value of COM was simulated in SFE and CSE, considering four different scenarios, both for the industrial and laboratory scale, considering the sale of by-products such as saponins and oil. Additionally, a linear analysis was carried out to evaluate the influence of two input variables (productivity and hydrolysate yield) on two economic indicators: cost of manufacturing (COM) and net present value (NPV).

In addition to COM, to carry out the sensitivity analysis, the gross margin (GM), return over the investment (ROI), payback time (PBT), internal rate of return (IRR), and net present value (NPV) at 7% interest were also simulated considering the above-mentioned selling prices of oil, saponins, and quinoa protein hydrolysate.

### 2.7. Statistical Analysis

All measurements were performed in triplicate. The data were expressed as mean ± standard deviation, and SPSS 24.0 was used for statistical analysis (Tukey test, *p* < 0.05, and Student’s *t*-test, *p* < 0.05). For the significance of NPV, the non-parametric Kruskal–Wallis test for independent samples was applied, followed by the Dunn–Bonferroni pairwise comparisons test, and *p* < 0.05 was considered significant. For the evaluation of the statistical correlation between the input variables (QPH yield and productivity of QPH) on economic indicators (COM and NPV), a multifactorial equation was obtained using linear regression.

## 3. Results and Discussion

### 3.1. Experimental Results

Optimization studies of oil extraction from quinoa flour with SFE using CO_2_ were performed, and the quinoa flour had an oil content of 7.0%. The seeds were ground with a mill IKA Basic 11, to obtain particle size of ≤0.5 mm. The effect of the cosolvent on the efficiency of oil extraction from quinoa flour was studied by using ethanol in an amount of 10–92% (weight of the cosolvent with respect to weight of the quinoa flour). The best results were obtained for the cosolvent percentage (range up to 90%), as can be seen in Figure 4.

As shown in Table 4, SFE allowed to obtain a higher oil extraction yield (OEY) of 28.49% compared to CSE, and the results are in agreement with previous studies, where values obtained with SFE were higher than those defatted by Soxhlet, with values reported for SFE of approximately 89% of yield [4,14]. In terms of nutritional characteristics, in a previous study reported by Estrada et al. [33], the fatty acid profile obtained by SFE was as follows: C14:0 (0.31%), C16:0 (10.01%), C18:0 (0.68%), C18:1w-9 (25.53%), C18:1-w-7 (0.94%), C18:2w-16 (49.56%), and C18:3w-3 (3.69%), similar to that reported by Wejnerowska and Ciaciuch et al. [14]: C16.0 (11.2%), C18:1 (28.2%), and C18:2 (50.9%), although it is true that no significant differences were reported in the fatty acid profile of oil obtained by SFE and other conventional methods with organic solvents [4,14,19]. SFE extracted oil may contain significant amounts of tocopherols (vitamin E), according to previous studies [15]. Additionally, as can be seen in Table 4, SFE also allows the extraction of phenolic compounds in the defatted quinoa flour stage [19]. The phenolic compounds were recovered at 34.28 mg GAE/100 g, and similar studies report that the oil obtained by SFE of canola has a higher phenolic content [34], giving it functional antioxidant properties and improving the stability of the oil in terms of quality [15].

It is important to emphasize the high nutritional value of quinoa oil in terms of the main components in the oil, which were linoleic acid (~50%), oleic acid (~25%), and palmitic acid (~10%), making up a total of 78% of polyunsaturated fatty acids (oleic acid, linoleic acid, and linolenic acid), with the latter two being considered essential fatty acids, which have functionality and health properties which may help reduce the risk of various diseases [35], resulting in an oil with high antioxidant potential and a higher market value. The average cost of quinoa oil in the international market is US$ 33.5 L. Due to its high nutraceutical potential and to increase its productivity, it is desirable to extract it on an industrial scale. Additionally, this also confirms the feasibility of using SFE to obtain defatted quinoa as an intermediate product in food applications, free of solvent residues, and with a technological quality superior to that obtained by conventional organic solvents [36].

Studies have reported on flavonoid and phenolic acids of quinoa seeds carried out by HPLC, and the main phenolic acids found were gallic acid, p-hydroxybenzoic acid, vanillic acid, p-coumaric acid, caffeic acid, and cinnamic acid, but the main flavonoids found in the seeds were orientin and rutin [37]. In the present study, the total flavonoids content of QPH was analyzed by the colorimetric method, and a higher amount (86.62% more) of flavonoids expressed as mg RE/g was found in the sample with the CSE process than in the sample with SFE pretreatment, confirming that there are significant differences in the amounts of remaining flavonoids in QPH (*p* < 0.05).

Previous studies have shown that the addition of ethanol increases the extraction yield of flavonoids and other biological compounds, and these values are always higher when extracted with water [4,14]. In this sense, it can be mentioned that SFE allows the extraction of flavonoids from quinoa flour, achieving a higher degree of purification of the sample and leaving a minimum amount of these compounds in the quinoa protein hydrolysates [38]. The remaining PC in QPH with the SFE process allows a higher degree of purification of the quinoa flour, reducing it by 85.84%; to date, no similar work has been reported. The QPH yield with SFE was 22% higher than that obtained with CSE, and this may be due to the higher protein yield content reported in the previous research [19], and other studies also reported similar values [39,40]. Therefore, the high antioxidant capacity of the QPH pretreated with SFE is mainly due to the bioactive peptides and not to the phenolic compounds, as is the case in the sample pretreated with CSE. Finally, SFE allowed better overall QPH performance, which was 1.23 times higher compared to CSE (Table 4).

Additionally, the antioxidant activity value obtained for SFE in our previous study was 1181.64 μmol TE/g protein, and for CSE it was 1448.84 μmol TE/g protein [19]. Other similar studies report values of 102.9, 15.3, and 60.1 expressed as ascorbic acid µg equivalents/mL of hydrolysate by ABTS using commercial enzymes at 120 min [41]. This confirms the high antioxidant activity and functionality of QPH, and these have been extensively demonstrated in in vitro and in vivo studies, where dietary intake of antioxidants has been reported to be helpful in protecting the organism from oxidative stress and reducing the risk of chronic diseases related to oxidative stress [16]. On the other hand, from a nutritional point of view, essential amino acids have been found in quinoa protein, with the main ones being: lysine, leucine, valine, and histidine, which had the highest amounts of 11.7%, 6.73%, 4.17%, and 4.17%, superior to other common cereals [42,43]. These amino acids present in hydrolyzed peptides confer antioxidant functionality, such as free radical scavenging; for example, lysine and valine can scavenge hydroxyl radicals by acting as electron donors that generate a hydrophobic microenvironment in the molecule [44]. Mudgil et al. report a peptide found in quinoa hydrolysate (QHPHGLGALCAAPPST) with multifunctional properties beneficial to health [45].

### 3.2. Economic Evaluation of QPH Production

The CSE and SFE processes were simulated using SuperPro Designer 9.0^®^ software in order to determine the COM of the QPH production; for the base simulation, the sale of the two by-products (oil and saponins) was considered. The COM, productivity, and total capital investment were also obtained (Figure 5) for both process and production scales (1.5 and 2500 kg). For the SFE process, the COM of QPH based on the total amount obtained decreased 90 times, and for the CSE process, the COM decreased 77 times, as the production scale increased. Therefore, considering the plants with the highest production capacities amongst those simulated, the highest productivity and the lowest COM were obtained. In this sense, the 2500 kg plant was the most appropriate amongst those simulated for producing QPH (269,998 tons/year) from quinoa seeds by the SFE process with the lowest COM (US$ 26.33/kg), compared to the CSE process (57,734 tons/year) with the lowest COM (US$ 57.06/kg). TCI for CSE and SFE was approximately US$ 2,300,000.00 and US$ 32,000,000.00, respectively, and for both cases, these increased with the industrial scale.

The productivity (t/year) represents the sum of accumulated QPH obtained from each production batch (mass basis). The behaviors of COM, productivity, and TCI for different plant capacities are also corroborated by other studies, such as passion fruit extract [27], curcuminoid powdered extract obtained from turmeric [46], and sucupira oil from sucupira branca seeds [47], all of which applied supercritical technology.

Figure 6 and Figure 7 show the contributions of the main cost factors (CRM, COL, FCI, CUT, and CWT) on COM for each extraction process. For SFE, the FCI and COL were the components that presented the highest contribution to COM at the laboratory scale, but CRM and FCI were the components that presented the highest contribution to COM at the industrial scale. The increase in CRM with scale-up is related to the increase in the required amount of raw material (1.5 kg/batch quinoa seeds to 2500 kg/batch quinoa seeds), and the largest contribution from the CRM was expected, since the raw material has a considerable cost (US$ 1.57/kg). A similar result was obtained in other simulations considering short process times for SFE [39]: FCI decreased with the increase in scale from 40.44% to 28.92%, and this decrease is not considerable because the investment is high, mainly due to the cost of the extraction unit with supercritical fluids which amounts to US$ 9,828,054.25. In the case of CSE, the FCI and COL were the components that presented the highest contribution to COM at the laboratory scale, but CRM and COL were the components that presented the highest contribution to COM at the industrial scale. The increase in CRM with scale-up is related to the increase in the required amount of raw material. COL decreased with the increase in scale from 51.23% to 19.12%, and this decrease is not considerable because the total operation time, including handling, loading, and unloading of the solvent (petroleum ether), requires 19 h per batch, and a large extraction time results in a minor number of batches per year and, consequently, a greater amount of man-hours [48].

However, when we compared the two processes on an industrial scale, FCI contributes more in SFE than CSE, the FCI was 164 times higher for SFE compared to CSE. Due to the high cost of the extraction unit with supercritical fluids, in both cases, the cost of the material contributes significantly to the COM, however, CRM impacts 1.2 times more on SFE compared to CSE, due to the shorter extraction time of 4 h compared to 19 h. Finally, the COL value decreases substantially when industrializing the process using both methods, because despite needing the same number of people considering three shifts, the proportion of other elements of the manufacturing cost are higher, so the percentage of this factor with respect to the others decreases. A similar result was obtained in other simulations [27].

### 3.3. Sensitivity Analysis and Comparison between Extraction Methods

The sensitivity analysis was performed considering four scenarios. Scenario 1 is the base case, including the sale of QPH and the sale of the two by-products (oil and saponins), scenario 2 is the sale of QPH and only the sale of saponins, scenario 3 is the sale of QPH and only the sale of oil, and scenario 4 is the sale of QPH without the sale of by-products. The four scenarios were considered, both at the laboratory and industrial scale, and for each type of process (SFE and CSE), and the results are shown in Table 5.

As shown in Table 5, when the CSE was used, considering the four different scenarios both at the laboratory and industrial scale, the COM of one kg of QPH ranged between US$ 4367.18 and US$ 4428.38 for the laboratory scale, and for industrial scale ranged between US$ 57.06 and US$ 109.29. At the laboratory scale, the main factor impacting on the COM was the COL, which impacts on approximately 51.23%, but at the industrial scale it was CRM, which impacts on 55.54%. This occurs with the increase in capacity: a higher production rate of QPH will require a higher consumption of quinoa seeds, and the same was observed in previous studies [46,49]. The best scenario on an industrial scale occurs when the sale of QPH and by-products such as oil and saponins takes place, decreasing the value of COM by approximately 1.92 times without the sale of by-products, and the parameters of return indicate the feasibility of the process. For instance, the GM and ROI were 42.40% and 155.83%, respectively. The PBT was 0.64 years with an NPV of US$ 28,159,000.00.

In the same trend, when the SFE was performed, two by-products were obtained in each batch. COM of one kg of QPH ranged from US$ 2599.68 to US$ 2660.88 at the laboratory scale, but at the industrial scale ranged from US$ 28.90 to US$ 90.10. In general, COM values are a bit higher in CSE than SFE because the production rate of QPH, and hence the productivity, increased significantly. For example, on an industrial scale, the productivity of SFE was 269,998 tons/year, compared to CSE of 57,734 tons/year. For this itemized cost, FCI is contributing to approximately 28.92% of the total cost, being 7.45 times higher compared to CSE. In this process, both CRM and FCI significantly impact on the COM. The COM values of SFE were lower than CSE in all four scenarios; therefore, all financial indicators were also lower, and all scenarios for SFE presented a positive return to the initial capital and operational investment. The best scenario for processing QPH by SFE was achieved by integrating SFE-I-1, which decreased the COM by 3.12 times with respect to the scenario SFE-I-4. In such scenario, the GM and ROI were 67.31% and 85.96%, respectively. The PBT was 1.16 years with an NPV of US$ 205,006,000 (Table 5).

As shown in Table 5, the COM calculated at the industrial scale was lower that the selling price of QPH (US$ 200/kg) in all four scenarios with the two technologies (CSE-I-1, CSE-I-2, CSE-I-3, CSE-I-4, SFE-I-1, SFE-I-2, SFE-I-3, SFE-I-4), which suggests that both production processes are profitable under those conditions. Raw material can represent up to 80% of the COM when SFE is used [34]. According to Osorio-Tobón et al. [46], raw materials, despite their high variability in cost, are generally the components with the highest contribution to COM. In the present study, when scenarios SFE-I-1, SFE-I-2, SFE-I-3, and SFE-I-4 were assessed, the COM was 1.21 times higher on SFE compared to CSE scenarios CSE-I-1, CSE-I-2, CSE-I-3, and CSE-I-4; however, using the first process, CMR had a major impact on the COM. In the latter scenario, our COM for SFE was lower than the COM obtained for the defatting of annatto seeds using supercritical carbon dioxide as a pretreatment for the production of bixin [50]. Other studies also corroborate the superiority of SFE over conventional oil extraction technologies, with better technological and economical results [34].

GM evaluates the short-term benefits of the extraction process [51]: a higher GM indicates that the project is more feasible because this indicator represents the percentage of every dollar of a product sold that the company retains as gross profit [29,51]. In general, for both processes of QPH production, the GM was negative in all lab-scale scenarios (CSE-L-1, CSE-L-2, CSE-L-3, CSE-L-4, SFE-L-1, SFE-L-2, SFE-L-3, SFE-L-4) because the COM was higher than the selling price of QPH; however, at the industrial scale, the COM decreased, observing a higher GM in all industrial-scale scenarios (CSE-I-1, CSE-I-2, CSE-I-3, CSE-I-4, SFE-I-1, SFE-I-2, SFE-I-3, SFE-I-4). Similar results were found in [27], which obtained positive GM values for the extraction from passion fruit by-products, at a selling price higher than US$ 200/kg. On the other hand, in the SFE, high GM values were obtained during all the evaluated scenarios. However, in the most favorable scenario (SFE-I-1), the GM was significantly higher than that reported for integrated annatto seeds-sugarcane biorefinery using supercritical CO_2_ extraction as a first step at a GM of 49.3% [52].

Another important parameter to evaluate the performance of the extraction processes is the ROI, and the higher this value, the more attractive the project [53]. However, for a project to be feasible, a minimum ROI between 10% and 15% is acceptable [51]. Similar to GM, in both extraction processes, ROI increased as COM decreased, reaching its highest value in both extraction processes with scenarios SFE-I-1 and CSE-I-1. When comparing both processes, it was observed that in the case of the CSE, the ROI was more influenced by the costs of the raw material and the cost of labor, while in the SFE, CRM and costs of the plant (FCI) had a higher impact on the ROI due to a higher investment in equipment to carry out this process. ROI values higher than 80% were reported as indicators of good profitability for a pulp oil production plant’s operation from Caryocar Brasiliense using supercritical technology [54]. An attractive GM of 67.31% was found in scenario SFE-I-1, which corresponds to the highest ROI. This GM value means that approximately 67.31% of the revenue from QHP and by-product sales would come in as gross profit to the company. In the pequi oil extraction project, a value close to the GM of 75% was reported with a projected sales price of US$ 150.00/kg of oil and productivity of 890,462 kg oil/year [54].

PBT is also an important parameter in the sensitivity analysis, which allows to assess the time to recover the initial investment. It is estimated that the shorter the PBT, the faster the initial investment will be recovered; however, this depends on the type of company and the investors [55]. For small and large plants, the PBT should be between 2 to 3 years and 7 to 10 years, respectively [51]. In the case of the SFE, a time between 1.16 to 1.43 years was obtained when the COM ranged from US$ 28.90 to US$ 90.10, which would indicate the feasibility of the process during all the evaluated scenarios. When scenarios SFE-I-1, SFE-I-2, SFE-I-3, and SFE-I-4 were compared with the CSE scenarios CSE-I-1, CSE-I-2, CSE-I-3, and CSE-I-4, they were higher by 1.81, 1.48, 1.80, and 1.46 times, respectively, due to a higher investment in equipment in SFE. A capital recovery time ranging from 0.6 to 1.5 years was calculated in the project where annatto seeds were processed using a 100 L extraction plant [56]. A longer capital recovery time is consistent with the current findings because larger extraction units require higher capital investment.

Finally, the NPV assesses the present value of all future cash flows generated by a project, including the initial capital investment, allowing to establish which projects could generate the most profit [55]. A project can be considered feasible if the NPV is positive after generally assuming an interest rate of 7% [46,57]. For both processes, all scenarios (CSE-L-1, CSE-L-2, CSE-L-3, CSE-L-4, SFE-L-1, SFE-L-2, SFE-L-3, SFE-L-4) at the laboratory scale were negative, because all COMs were very high. At the industrial scale, in the CSE, all scenarios (CSE-I-1, CSE-I-2, CSE-I-3, CSE-I-4) had a high NPV ranging from US$ 17,671,000 to US$ 28,159,000; however, in the SFE, all the evaluated scenarios (SFE-I-1, SFE-I-2, SFE-I-3, SFE-I-4) present a positive NPV value, and these were higher compared to the CSE scenarios, by 7.28, 8.96, 7.34, and 9.1 times, respectively, which indicates that all of them are better. Whenever CSE did not consider the utilization of all by-products, this difference was much higher, and due to the higher productivity of SFE compared to CSE, the higher production flow from QPH allows for a higher sales volume and therefore higher net revenue streams [27]. However, to perform a more adequate economic evaluation of the cost of production of QPH by the pretreatment methods evaluated in the current study, other factors such as raw material characteristics and seasonality, market size, product demand, and costs related to product quality control, packaging, and distribution must also be considered [49]. A supercritical plant project’s profitability must seek a balance between technical and economic criteria, aiming at higher profitability [58].

### 3.4. Statistical Analysis

When assessing statistical correlation between the two input variables considered (QPH yield and productivity of QPH production plant), on two important economic indicators of the process (COM and NPV), a multifactorial equation was obtained for each of the two indicators evaluated, as shown in Equations (4) and (5) below:COM = 53,473.63 − 32,502.49 (QPH yield)(4)
NPV = −6,608,839.99 + 693.72 (Productivity of QPH)(5)

The linear regression equations were obtained by discarding the one with the lowest significance in the model, which represents a dependent variable. The excluded variable for the COM was productivity; however, for the NPV variable, the excluded variable was hydrolysate yield. The results obtained show that the COM maintains a strong relationship with the variable QPH yield and the variable productivity of QPH. These models allow for estimating the main dynamic indicators of the investment for different variations in hydrolysate yield and productivity. In this way, QPH producers can make predictions of the economic behavior of the plant for different scenarios of the two initial variables considered.

The significance of the type of pretreatment (SFE, CSE) was evaluated for each production scale on the COM, and the results are shown in Table 6. The type of pretreatment showed a significant effect on COM of QPH at the laboratory scale (*p* < 0.05); however, the type of pretreatment did not have a significant effect on the COM at the industrial scale. Although no significant differences were found, it can be affirmed that pretreatment with SFE allowed to obtain a lower COM of QPH compared to CSE. The shorter extraction time in SFE allows for a higher QPH production rate and higher productivity [50]; however, this is counterbalanced by a higher FCI and TCI, and therefore for the mean of the four scenarios, no significant differences were observed [27]. Accordingly, it is possible to observe that plant cost was the most sensitive parameter [59].

Many studies have demonstrated the superiority of SFE over conventional extraction methods; in that sense, according to the findings for both processes (SFE and CSE), all four industrial scale scenarios proved to be techno-economically viable. It is important to note that the present work is a preliminary study, and when the process remains promising and the scale-up stage is successfully passed, the process will finally reach the point of being studied under deeper economic evaluations [58].

## 4. Conclusions

The SFE process showed better results than CSE: the OEY was 29.49% higher, the remaining TFCs were reduced by 85.84%, and the QPH yield was 1.23 times higher, and this confirms the higher degree of purification of QPH, where the high antioxidant activity mainly comes from the bioactive peptides and not from other bioactive compounds. Additionally, the SFE oil also presented high antioxidant capacity, improving its nutritional and functional quality and its stability, as well as in QPH and oil, both cases free of organic solvent contaminants. Scaling up increased the process productivity and decreased the COM for both proposed processes. The significance analysis of the factors considered showed that there was no significant effect on the COM and NPV of the QPH production at the industrial scale between each technology; however, the pretreatment with SFE allowed for obtaining a lower COM and a higher NPV. The sensitivity analysis and the evaluated scenarios showed an additional income generated by the sale of by-products such as oil and saponins. Generally, it can be concluded that hydrolysates from quinoa have potential to be used as functional ingredients or nutraceuticals to prevent oxidative cell damage of free radicals or to be incorporated as additives to improve food preservation. Oil from quinoa can be used in cosmetic, pharmaceutical, and food industries, which favorably contribute to human health and nutrition. Since it is economically feasible to produce them on an industrial scale using both technologies, this study constitutes the basis and can complement with others studies in order to carry out the industrialization.

## Figures and Tables

**Figure 1 foods-11-01015-f001:**
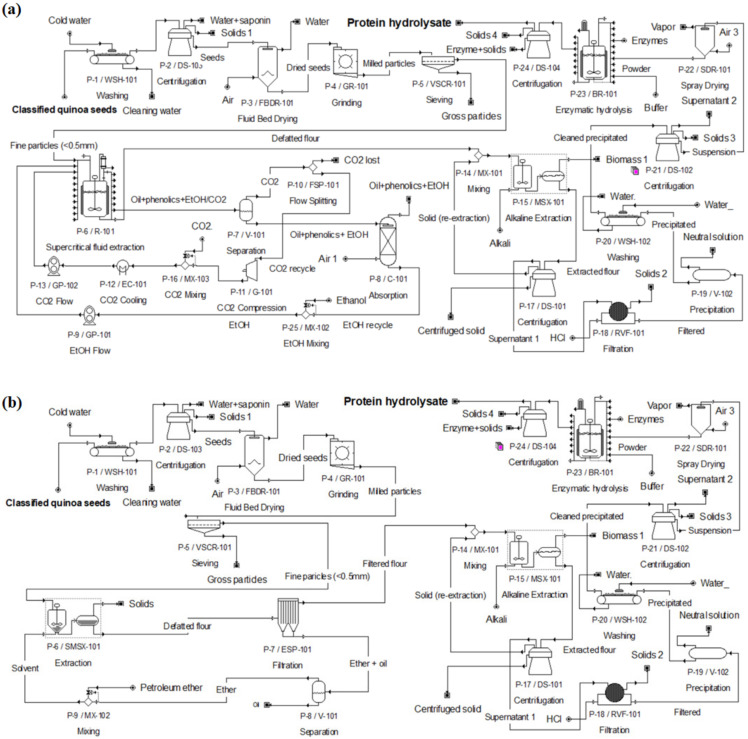
Flowsheets of the (**a**) supercritical fluids’ extraction (SFE) and (**b**) conventional solvent extraction (CSE) simulated with SuperPro Designer 9^®^ for QPH production. Source: [18], reproduced with permission from Olivera-Montenegro et al., The 2nd International Electronic Conference on Foods 2021—Future Foods and Food Technologies for a Sustainable World, sciforum-048913; published by MDPI, 2021.

**Figure 2 foods-11-01015-f002:**
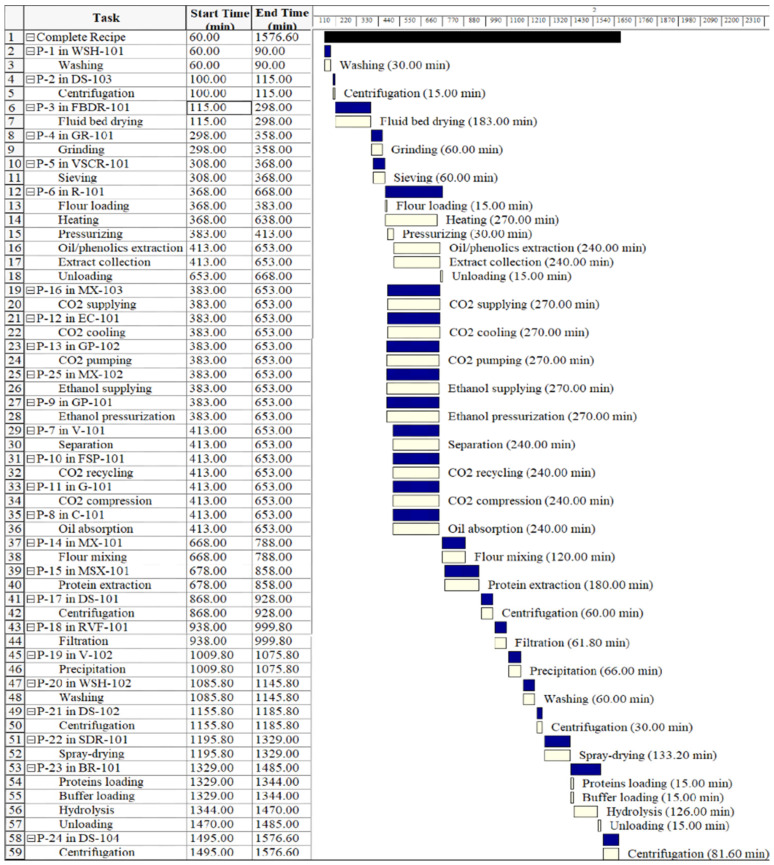
Industrial operations Gantt chart for obtaining QPH by supercritical fluids’ extraction (SFE).

**Figure 3 foods-11-01015-f003:**
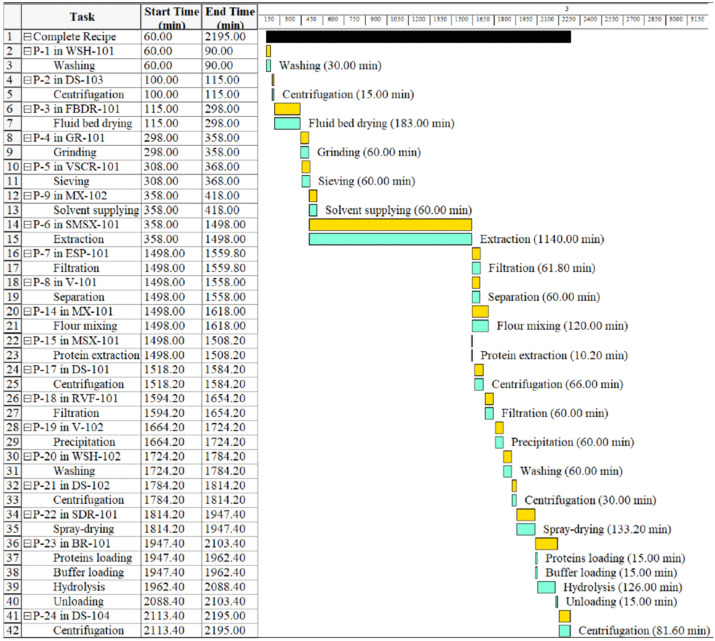
Industrial operations Gantt chart for obtaining QPH by conventional solvent extraction (CSE).

**Figure 4 foods-11-01015-f004:**
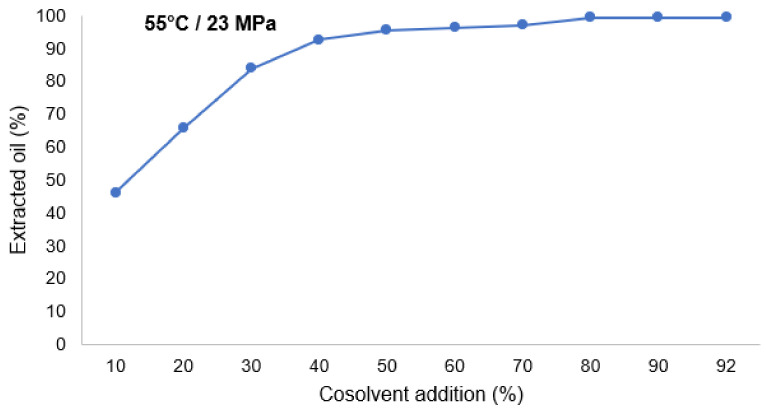
Effect of ethanol cosolvent addition (%) on the extracted oil (extraction of quinoa flour with supercritical CO_2_ at 23 MPa and 55 °C).

**Figure 5 foods-11-01015-f005:**
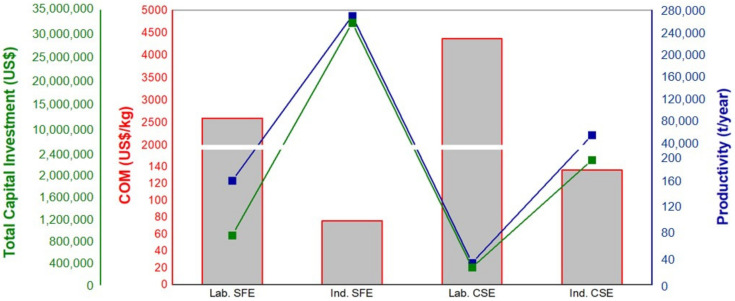
COM, productivity, and total capital investment to produce QPH using SFE and CSE (Lab. SFE: Laboratory supercritical fluids’ extraction; Ind. SFE: Industrial supercritical fluids’ extraction; Lab. CSE: Laboratory conventional solvent extraction; Ind. CSE: Industrial conventional solvent extraction).

**Figure 6 foods-11-01015-f006:**
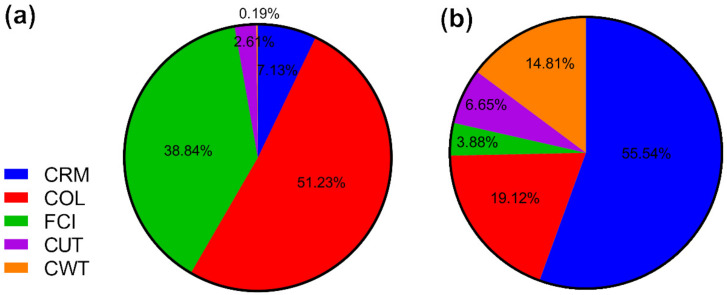
Contribution of each component. CRM: cost of raw material; FCI: fixed capital investment; COL: cost of operational labor; CUT: cost of utilities on the COM for quinoa protein hydrolysate obtained by supercritical fluids; extraction (SFE) for (**a**) laboratory scale and (**b**) industrial scale.

**Figure 7 foods-11-01015-f007:**
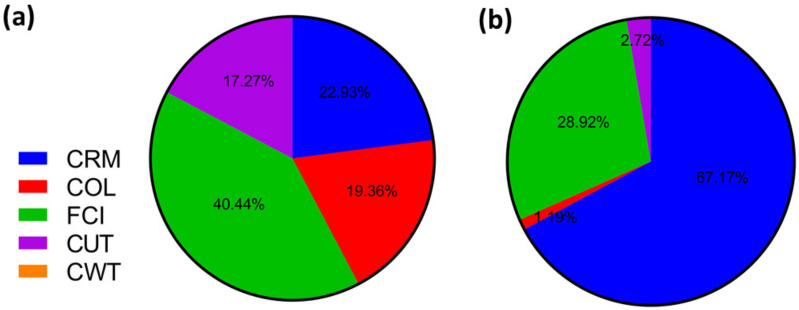
Contribution of each component. CRM: cost of raw material; FCI: fixed capital investment; COL: cost of operational labor; CUT: cost of utilities on the COM for quinoa protein hydrolysate obtained by conventional solvent extraction (CSE) for (**a**) laboratory scale and (**b**) industrial scale.

**Table 1 foods-11-01015-t001:** Experimental data used to simulate the production of QPH by supercritical fluids’ extraction (SFE) and conventional solvent extraction (CSE).

Parameter	CSE	SFE
Washing—1st step		
Saponin extraction yield	0.31 g saponins/100 g quinoa seeds (db) ^a^	0.31 g saponins/100 g quinoa seeds (db) ^a^
S/F	15 L water/1.5 kg quinoa	15 L water/1.5 kg quinoa
Temperature (Washing 1)	50 °C × 30 min	50 °C × 30 min
Temperature (Washing 2)	20 °C × 30 min	20 °C × 30 min
Extraction time	1 h	1 h
Defatted—2nd step		
Oil extraction yield	4.58 g fat/100 g quinoa flour (db)	6.30 g fat/100 g quinoa flour (db)
Temperature	50 °C	55 °C
Time	19 h	4 h
S/F	1000 mL petroleum ether/300 g quinoa flour	100 mL ethanol/8 g quinoa flour
CO_2_ flow rate	-	35 g CO_2_/min
Pressure	-	23 MPa
Bulk density	0.55 g/mL ^b^	0.55 g/mL ^b^
Protein extraction—3rd step		
Extraction yield	11.74 g protein/100 g quinoa (db)	12.78 g protein/100 g quinoa (db)
Time	2 h	2 h
Temperature	120 °C	120 °C
Spray drying—4th step		
Extraction yield	5.57 g protein concentrate/100 g quinoa (db)	6.84 g protein concentrate/100 g quinoa (db)
Time	2.22 h	2.22 h
Temperature		
Enzymatic hydrolysis—5th step		
QPH yield	160.52 g hydrolysate/100 g quinoa	197.12 g hydrolysate/100 g quinoa
Time	2 h	2 h
Enzyme concentration	4.2 UHb ^c^/g protein	4.2 UHb ^c^/g protein
pH	7	7

S/F: mass ratio of solvent to feed. ^a^ Calculated based on [17,18]. ^b^ Calculated based on [25]. ^c^ UHb/g: Units of hemoglobin/g.

**Table 2 foods-11-01015-t002:** Base cost for each plant for production of QPH using supercritical fluids’ extraction (SFE) and conventional solvent extraction (CSE).

Stage	Equipment	*n* ^a^	Unit Base Cost (US$)	CSE		SFE	
				Number of Equipment	Total Cost (US$)	Number of Equipment	Total Cost (US$)
Selection	Optic sorting system	0.89	7000.00	1	7000.00	1	7000.00
Washing	Stirred tank reactor (10 m^3^) ^c^	0.45	20,000.00	3	60,000.00	3	60,000.00
Saponin extract concentration	Evaporator ^c^	0.59	100,000.00	1	100,000.00	1	100,000.00
Drying of saponins	Spray dryer ^c^	0.59	97,000.00	1	97,000.00	1	97,000.00
Centrifugation	Decanter (3 m^3^/h) ^c^	0.71	4500.00	1	4500.00	1	4500.00
Drying	Fluid bed dryer ^c^	0.65	56,000.00	1	56,000.00	1	56,000.00
Grinding and sieving	Hammer mill ^c^	0.91	8000.00	1	8000.00	1	8000.00
Defatting	Stirred tank reactor with cooling system (15 m^3^) ^c^	0.45	30,000.00	2	60,000.00	-	-
	Supercritical fluids unit (4 m^3^) ^b^	0.60	154,764.16	-	-	1	9,828,054.25
Separation (oil-ethanol)	Rotary Evaporator ^c^	0.49	16,918.42	1	499,298.00	1	499,298.00
Alkaline extraction	Stirred tank reactor (2 tanks 10 m^3^ and 1 tank 5 m^3^) ^c^	0.45	15,000.00	3	45,000.00	3	45,000.00
Centrifugation 2	Decanter (20 m^3^/h) ^c^	0.71	30,000.00	1	30,000.00	1	30,000.00
Filtration	Filter press ^c^	0.66	6077.00	3	18,231.00	3	18,231.00
Precipitation	Stirred tank reactor ^c^	0.45	21,000.00	2	42,000.00	2	42,000.00
Centrifugation 3	Decanter (20 m^3^/h) ^c^	0.71	30,000.00	1	30,000.00	1	30,000.00
Washing 2	Washing tank (2.5 m^3^) ^c^	0.53	5000.00	1	5000.00	1	5000.00
Centrifugation 4	Decanter (3 m^3^/h) ^c^	0.71	4500.00	1	4500.00	1	4500.00
Neutralization	Stirred tank reactor (0.5 m^3^) ^c^	0.45	1000.00	1	1000.00	1	1000.00
Spraying	Industrial spray dryer ^c^	0.59	75,000.00	1	75,000.00	1	75,000.00
Enzymatic hydrolysis	Stirred tank reactor (2.5 m^3^) ^c^	0.45	5000.00	2	10,000.00	2	10,000.00
Centrifugation	Decanter (5 m^3^/h) ^c^	0.71	7500.00	1	7500.00	1	7500.00
Transport of total solids	Trough conveyor ^c^	0.89	441.00	7	3087.00	7	3087.00
Transport of total fluids	Pump ^c^	0.55	2577.00	11	28,347.00	11	28,347.00
Total					1,299,194.00		10,968,248.25

^a^*n* constant depending on equipment type based on [28,31,32]. ^b^ Calculated based on [30]. ^c^ Direct quotation.

**Table 3 foods-11-01015-t003:** Input economic parameters used for COM simulation.

	Laboratory Scale (1.5 kg/batch)	Industrial Scale (2500 kg/batch)
Fixed capital investment (FCI) ^a^		
Conventional solvent extraction	$203,532.49	$1,200,194.00
Supercritical fluids’ extraction ^b^	$358,668.76	$10,968,248.25
Depreciation rate	10%/year	10%/year
Annual maintenance rate	6%/year	6%/year
Cost of operational labor (COL)		
Wage ($/hour) $ ^a^	$2.34	$2.34
Number of workers per shift	2	6
Cost of raw material (CRM)		
Quinoa seeds ^a^	1567$/tons	1567$/tons
Industrial CO_2_ ^a^	0.033$/kg	0.033$/kg
Absolute ethanol ^a^	0.53$/kg	0.53$/kg
Petroleum ether ^a^	859$/tons	859$/tons
NaOH 1 N ^a^	125$/tons	125$/tons
HCl 1 N ^a^	41.37$/tons	41.37$/tons
NaOH 0.1N	120$/tons	120$/tons
Phosphate buffer	1160$/tons	1160$/tons
Endopeptidase COROLASE^®^ 7089	27.73$	27.73$
Cost of utilities (COU)		
Electricity	0.1183$/kw	0.1183$/kw
Water	1.63$/ton	1.63$/ton
Cost of waste treatment (CWT)	100$/ton	100$/ton

^a^ Based on local quotations. ^b^ Calculated based on [30]. Source: [18], reproduced with permission from Olivera-Montenegro et al., The 2nd International Electronic Conference on Foods 2021—Future Foods and Food Technologies for a Sustainable World, sciforum-048913; published by MDPI, 2021.

**Table 4 foods-11-01015-t004:** Quinoa protein hydrolysate (QPH) yield, total flavonoid content, total phenolic content, and oil extraction yield obtained by conventional solvent extraction (CSE) and supercritical fluids’ extraction (SFE).

	CSE	SFE	Reference
QPH yield (%)	160.52% (hydrolyzed)	197.12% (hydrolyzed)	-
TFC (μg RE/g)	113.22 ± 8.13	15.97 * ± 1.17	-
TPC (mg GAE/100 g)	ND	34.28	Estrada et al. [33]
OEY (%)	77.59% (extracted oil)	99.70% (extracted oil)	Olivera-Montenegro et al. [19]

GAE: Gallic acid equivalent (GAE); OEY: Oil extraction yield; TFC: Total flavonoid content; ND: not detected; RE: Rutin equivalent; TPC: Total phenolics content. * Student’s *t*-test for independent samples, *p* < 0.05.

**Table 5 foods-11-01015-t005:** Cost of manufacturing of quinoa protein hydrolysate (QPH) for both scales (laboratory = 1.5 kg/batch and industrial = 2500 kg/batch).

Process Plant Scenario	Sale of Saponins	Sale of Oil	Productivity (tons/year)	COM (US$/kg)	CRM (%)	COL (%)	FCI (%)	CUT (%)	CWT (%)	GM (%)	ROI (%)	PBT (year)	NPV (at 7% Interest) (US$)	Operating Cost (US$/year)	Revenues (US$/year)
SFE-L-1	Yes	Yes	162	2599.68	22.93	19.36	40.44	17.27	0.00	−1019.12	−33.86	-	−3,470,000	421,000.00	37,000
SFE-L-2	Yes	No	162	2641.68	22.93	19.36	40.44	17.27	0.00	−1190.85	−34.40	-	−3,512,000	421,000.00	32,000
SFE-L-3	No	Yes	162	2618.79	22.93	19.36	40.44	17.27	0.00	−1025.88	−33.88	-	−3,472,000	421,000.00	37,000
SFE-L-4	No	No	162	2660.88	22.93	19.36	40.44	17.27	0.00	−1199.84	−34.43	-	−3,514,000	421,000.00	32,000
CSE-L-1	Yes	Yes	35	4367.18	7.13	51.23	38.84	2.61	0.19	−1751.31	−36.63	-	−1,305,000	151,000.00	7000
CSE-L-2	Yes	No	35	4409.26	7.13	51.23	38.84	2.61	0.19	−2065.05	−36.99	-	−1,315,000	151,000.00	6000
CSE-L-3	No	Yes	35	4386.29	7.13	51.23	38.84	2.61	0.19	−1764.85	−36.64	-	−1,305,000	151,000.00	7000
CSE-L-4	No	No	35	4428.38	7.13	51.23	38.84	2.61	0.19	−2089.35	−37.01	-	−1,315,000	151,000.00	6000
SFE-I-1	Yes	Yes	269,998	28.90	67.17	1.19	28.92	2.72	0.00	67.31	85.96	1.16	205,006,000	20,504,000	62,719,000
SFE-I-2	Yes	No	269,998	70.98	67.17	1.19	28.92	2.72	0.00	62.29	70.51	1.42	162,784,000	20,504,000	54,376,000
SFE-I-3	No	Yes	269,998	48.01	67.17	1.19	28.92	2.72	0.00	67.11	85.26	1.17	203,102.000	20,504,000	62,343,000
SFE-I-4	No	No	269,998	90.10	67.17	1.19	28.92	2.72	0.00	62.03	69.82	1.43	160,880,000	20,504,000	53,999,000
CSE-I-1	Yes	Yes	57,734	57.06	55.54	19.12	3.88	6.65	14.80	42.40	155.83	0.64	28,159,000	7,845,000	13,620,000
CSE-I-2	Yes	No	57,734	92.79	55.54	19.12	3.88	6.65	14.80	32.64	104.53	0.96	18,171,000	7,845,000	11,646,000
CSE-I-3	No	Yes	57,734	73.55	55.54	19.12	3.88	6.65	14.80	41.98	153.26	0.65	27,658,000	7,845,000	13,521,000
CSE-I-4	No	No	57,734	109.29	55.54	19.12	3.88	6.65	14.80	32.06	101.96	0.98	17,671,000	7,845,000	11,547,000

L: Laboratory; I = Industrial; SFE: Supercritical fluids’ extraction; CSE: Conventional solvent extraction; COM: Cost of manufacturing; CRM: Cost of raw material; FCI: Fixed cost of investment; CUT: Cost of utilities; CWT: Cost of waste treatment, GM: Gross margin; ROI: Return of investment; PBT = Payback time; NPV = Net present value. Source: [18], reproduced with permission from Olivera-Montenegro et al., The 2nd International Electronic Conference on Foods 2021—Future Foods and Food Technologies for a Sustainable World, sciforum-048913; published by MDPI, 2021.

**Table 6 foods-11-01015-t006:** Statistical analysis of the significance of the cost of manufacturing (COM) and net present value (NPV).

	Laboratory CSE	Laboratory SFE	Industrial CSE	Industrial SFE
COM * (US$/kg)	2630.26 ± 26.66 ^a^	4397.98 ± 26.69 ^b^	59.47 ± 26.69 ^c^	83.17 ± 22.72 ^c^
NPV ** (US$)	−3,492,000 ^ab^ 2.50	−1,310,000 ^a^ 6.50	182,943,000 ^ab^ 14.50	22,914,500 ^b^ 14.50

* Means with different subscripts differ at the *p* < 0.05 level, followed by Tukey’s Honest Significant Difference method. ** Medians with different subscripts differ at the *p* < 0.05 level, followed by Dunn–Bonferroni pairwise comparisons test.

## Data Availability

The data presented in this study are available upon request from the corresponding author.

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
