# Peer review of "Production of Protein Hydrolysate from Quinoa (Chenopodium quinoa Willd.): Economic and Experimental Evaluation of Two Pretreatments Using Supercritical Fluids’ Extraction and Conventional Solvent Extractionâ€"

_foods, 2022, doi:10.3390/foods11071015_

Round 1
Reviewer 1 Report
The search for alternative protein sources is one of the most important challenges today. Not only because of the need to reduce animal protein intake but also to diversify plant protein sources. Furthermore, a sustainable approach in food production is now a priority.
This paper addresses the effect of the extraction method on the yield and selected properties of the obtained quinoa protein hydrolysates. In the experiment, one of the most modern extraction methods - supercritical fluid extraction - was used, which the effects of which was related to the traditional solvent extraction.
The layout of the experiment was planned correctly, appropriate research methods were used and the results obtained were thoroughly discussed. In addition, the authors performed a thorough cost analysis of the implementation of the studied extraction methods considering both laboratory and industrial scale. This is very important especially when using modern and very expensive technologies such as supercritical fluid extraction.
Author Response
We agree with the reviewer, the research presented two technologies as pre-treatment for obtaining protein hydrolysate from quinoa, showing the superiority of SFE with respect to conventional technologies in terms of quality and economic aspect of the products obtained from quinoa as an alternative new source of protein. The feasibility of implementing the two processes on an industrial scale was thoroughly analyzed.
Reviewer 2 Report
The manuscript submitted by Olivera-Montenegro et al. addresses " Production of protein hydrolysate from quinoa (Chenopodium quinoa Willd.): Economic and experimental evaluation of two pretreatments using supercritical fluids extraction and conventional solvent extraction". This paper discusses a current topic, the importance to extract bioactive compounds and proteins from biomass. The article demonstrates the valorization of food since the extract obtained showed potential for further application in the food industry, for example. The economic comparison between the two extraction methods made by the authors is a very interesting approach since this kind of study allows us to understand well the advantages and disadvantages of each of the developed methods in economic aspects. The economic (and environmental) studies of the developed processes should be carried out more often, since these studies allow closer the university-industry relationship. I enjoyed reading the article, I think the ideas are well structured and linked.
- The authors should describe in the procedure which solvent they used for washing the quinoa and its temperature.
- The procedure should also state what type of equipment was used for conventional extraction, and why they chose that solid-liquid ratio as well as the extraction time. Note that time is one of the conditioning factors in conventional extraction, so the choice of 19h extraction has to be explained and justified.
- Line 128, why did you put a reference?
- In table 2 a dash is missing.
Author Response
Point 1: The authors should describe in the procedure which solvent they used for washing the quinoa and its temperature.
Response 1: In line 123 (word file), the changes were made as suggested, the solvent used, and temperature are described.
Point 2: The procedure should also state what type of equipment was used for conventional extraction.
Response 2: The type of equipment was used is a stirred-tank reactor, and this is specified in line 232 (word file). Additionally, the term “cooling system” is also incorporated to maintain the temperature of 4 °C for the extraction. Also updated in table 2 the term “cooling system”.
Point 3: and why they chose that solid-liquid ratio as well as the extraction time. Note that time is one of the conditioning factors in conventional extraction, so the choice of 19h extraction has to be explained and justified.
Response 3: In line 131 (word file), the changes were made as suggested. The solid-liquid ratio and extraction time, they were established as previously described by (Olivera-Montenegro, Fritz).
Point 4: Line 128, why did you put a reference?
Response 4: In Line 150 (word file), the reference was removed.
Point 5: In table 2 a dash is missing.
Response 5: In table 2, a dash was added.
Reviewer 3 Report
The aim of the paper s to compare the production of quinoa protein hydrolysate using two technologies to extract the oil and separate the phenolic compounds prior to enzymatic hydrolysis: Supercritical fluids extraction and Conventional solvent extraction The introduction is well structured and supported by bibliographic references that emphasize the nutritional, health and functional aspects of quinoa and its extracts. However, these aspects are not considered in the discussion of the results and conclusions.
As regards the chemical-nutritional traits, the article reports little information, as it reports almost exclusively technological and economic aspects, perhaps not adequate for this issue.
Author Response
Point 1: The aim of the paper is to compare the production of quinoa protein hydrolysate using two technologies to extract the oil and separate the phenolic compounds prior to enzymatic hydrolysis: Supercritical fluids extraction and Conventional solvent extraction The introduction is well structured and supported by bibliographic references that emphasize the nutritional, health and functional aspects of quinoa and its extracts. However, these aspects are not considered in the discussion of the results and conclusions.
Response 1: We agreed with the reviewer. The nutritional, health and functional value of products and by-products obtained from quinoa was emphasized.
Discussion: In line 352 and 415 (word file), the changes were made as suggested.
Conclusions: In line 661 and 673 (word file), the changes were made as suggested.
Round 2
Reviewer 3 Report
No comments.
Thanks to the authors for applying the corrections requested in this form.